# A Rare Case of Spontaneous Arachnoid Cyst Rupture Presenting as Right Hemiplegia and Expressive Aphasia in a Pediatric Patient

**DOI:** 10.3390/children8020078

**Published:** 2021-01-24

**Authors:** Anne Bryden, Natalie Majors, Vinay Puri, Thomas Moriarty

**Affiliations:** 1Department of Neurology and Pediatrics, University of Louisville SOM, Louisville, KY 40202, USA; 2Department of Neurology and Pediatrics, Vanderbilt University, Nashville, TN 37420, USA; natalie.majors@vumc.org; 3Department of Neurology and Pediatrics, University of Louisville, Louisville, KY 40202, USA; vinay.puri@louisville.edu; 4Department of Neurological Surgery and Pediatrics, University of Louisville, Louisville, KY 40202, USA; tmoriarty210@gmail.com

**Keywords:** pediatrics, arachnoid cysts, spontaneous rupture, aphasia, neurological deficits

## Abstract

This study examines an 11-year-old boy with a known history of a large previously asymptomatic arachnoid cyst (AC) presenting with acute onset of right facial droop, hemiplegia, and expressive aphasia. Shortly after arrival to the emergency department, the patient exhibited complete resolution of right-sided hemiplegia but developed headache and had persistent word-finding difficulties. Prior to symptom onset while in class at school, there was an absence of reported jerking movements, headache, photophobia, fever, or trauma. At the time of neurology consultation, the physical exam showed mildly delayed cognitive processing but was otherwise unremarkable. The patient underwent MRI scanning of the brain, which revealed left convexity subdural hematohygroma and perirolandic cortex edema resulting from ruptured left frontoparietal AC. He was evaluated by neurosurgery and managed expectantly. He recovered uneventfully and was discharged two days after presentation remaining asymptomatic on subsequent outpatient visits. The family express concerns regarding increased anxiety and mild memory loss since hospitalization.

## 1. Introduction

Arachnoid cysts (ACs) are cerebral spinal fluid (CSF) collections between two layers of arachnoid membrane that are primarily congenital in origin or acquired secondary to trauma, infection, or skull-based surgery. ACs are a relatively common developmental anomaly that are frequently diagnosed incidentally by intracranial imaging in pediatric patients [1,2,3]. They represent 1% of all intracranial masses [4] with a prevalence of around 2.6% in children [1] and male predominance [1,5,6,7]. In general, arachnoid cysts have a benign course with a large proportion of patients remaining asymptomatic. However, they can be associated with clinical symptoms, such as chronic headache, seizure, asymmetric macrocrania, or hydrocephalus [8,9]. 

Arachnoid cysts may also present with symptoms as a consequence of cyst rupture causing acute subdural hygroma, subdural hematoma, and/or intracystic hemorrhage [2]. Larger arachnoid cyst size, a median size of 2.2 × 2.6 × 2.5 cm [5], and recent head trauma are risk factors for symptomatic rupture [10]. Spontaneous or nontraumatic rupture appear as a rare event, with a total of only 57 reported cases of spontaneous AC rupture [11]. The most frequently reported symptom following spontaneous rupture is headache (82%) and nausea/vomiting (35.3%). Neurological deficits and altered consciousness are rarely reported [11]. 

This case describes a rare spontaneous AC rupture, and, to our knowledge, this is the first spontaneous AC rupture that presents with acute onset focal neurological signs, namely facial nerve injury, expressive aphasia, and hemiplegia.

## 2. Patient Description 

The patient, an 11-year-old male with known history of a previously asymptomatic, left frontal arachnoid cyst, found incidentally on imaging after persistent headaches at age 4, was in his normal state of health until experiencing sudden-onset dizziness while in class. Shortly after, he developed a right-sided lower facial droop, right hemiplegia, and expressive aphasia. There was no history of trauma, seizures, or other neurological conditions, and the patient was otherwise in good health. These symptoms began to resolve en route to a pediatric tertiary care center via a medical stat flight. Upon arrival, the remaining symptoms included some right arm weakness, mild-to-moderate dysarthria, flattened nasolabial fold, and facial asymmetry on smiling. There was near-complete symptom resolution at the time of neurology consultation 5 min after initial exam and 70 min following symptom onset. The neurological exam showed mildly delayed cognitive processing, but the cranial nerve, motor, sensory, coordination, and gait exams were unremarkable. There was an absence of witnessed/reported convulsions, abnormal movements, loss of consciousness, headache, photophobia, or fever. There was no evidence of external trauma on physical exam or report of trauma preceding and/or during symptoms. 

Figure 1 and Figure 2 reveal a change in signal of arachnoid cyst suggestive of interval hemorrhage with a thin subdural hematoma along its inferior aspect. Minimal edema in the subjacent Rolandic cortex was present. Given the patient’s stroke-like presentation, an MRA of head and neck was administered, which was found to be normal. The findings, with a comparison of the previous MRI of the patient at age 4 (Figure 3 and Figure 4), are consistent with a ruptured arachnoid cyst. No acute neurosurgical intervention was performed due to the absence of raised intracranial pressure and reduction of neurological symptoms. A routine EEG performed in the emergency department exhibited interictal epileptiform discharges over the left and right central–temporal region during drowsy and sleep states, suggestive of increased risk for partial seizures from these regions.

The patient received 500 mg levetiracetam twice daily for seizure prophylaxis. On day 2 of admission, a rapid-sequence MRI brain scan showed a stable thin subdural hematoma and a ruptured arachnoid cyst at the left frontoparietal vertex with the interval resolution of edema in the subjacent cortex. At discharge on day 3, the patient’s neurological exam remained normal without development of symptoms or signs related to acute intracranial hypertension. The patient maintained on 500 mg levetiracetam BID for seizure prophylaxis for three months. A rapid-sequence MRI brain scan at 2-week outpatient follow-up revealed a mild interval decrease in the size of left convexity subdural hematohygroma. He has had no return of rupture symptoms or concerns for seizures, including nocturnal seizures. He has occasional headaches, not sustained or requiring medication. The family express concerns regarding occasional anxiety and mild memory loss since hospitalization. 

## 3. Discussion

ACs are CSF collections between layers of arachnoid membrane that typically present as asymptomatic and represent 1% of all intracranial masses [4]. In general, arachnoid cysts have a benign course with a large proportion of patients remaining asymptomatic. However, they can be associated with clinical symptoms, such as chronic headache, seizure, asymmetric macrocrania, or hydrocephalus. Head injury from recent trauma is the most common cause of AC rupture; rarely do they rupture spontaneously, reflected by the fact that there are only 57 reported cases of spontaneous AC rupture. In the case of traumatic rupture, these lesions are more associated with subdural hygroma, subdural hematoma, and/or intracystic hemorrhage, across all ages [2]. Different treatment options for rupture of ACs are very controversial. Immediate operative intervention, either through microsurgical fenestration or CSF derivation, is warranted by raised intracranial pressure and progressive neurological deterioration [11,12,13]. Conservative treatment has also been proposed depending on a patient’s intracranial pressure and neurological condition [9]. In the case of our patient, conservative treatment was favored due to only mild cerebral edema, no significant rise in intracranial pressure, and the reduction—and eventual cessation—of neurological symptoms. 

Our patient was diagnosed with non-traumatic rupture of left frontoparietal arachnoid cyst with associated subdural hematohygroma. This resulted in a transient right hemiplegia, right facial droop, and expressive aphasia. Presumably, these symptoms are secondary to the cyst rupture, release of proteinaceous fluid, and hemorrhage with subsequent inflammation. ACs have a prevalence of 2.6% in children [1], with a higher preponderance in males [1,5,6,7]. As reported in many studies, the main symptoms following spontaneous AC rupture are headache and nausea/vomiting [11,12], whereas focal neurological deficits are rare [2], this being the first case to our knowledge with hemiplegia and expressive aphasia. Our patient did not experience prolonged headaches or nausea/vomiting; instead, he presented with acute onset focal neurological symptoms. Additionally, our patient did not suffer from a traumatic event, and his AC ruptured spontaneously, which is extremely rare. The family concerns regarding increased anxiety and mild memory loss since hospitalization may be attributed to the possible side effect of levetiracetam [14].

Anticipatory guidance in pediatric patients diagnosed with arachnoid cysts is disputed among pediatric neurosurgeons. ACs are a relatively common developmental anomaly, which are frequently diagnosed incidentally by intracranial imaging in pediatric patients, leading most physicians to recommend conservative observational treatment [1,2,3]. When the patient is asymptomatic with a small AC, no treatment is the preferred course of action with periodic imaging to monitor the cyst’s growth. However, if symptoms arise or the cyst ruptures, the cyst will be re-evaluated and surgical treatment, via open craniotomy fenestration, endoscopic cyst fenestration, or cysto-peritoneal shunting [11], is recommended [1]. Larger arachnoid cyst size and recent head trauma are risk factors for symptomatic rupture. Mild head trauma is extremely common in children’s daily lives, and while the trauma can potentially lead to rupture, physicians believe that parents should be aware of this risk but not restrict sports or other activities for most children with ACs [10]. 

Discussion of risk factors, size of AC, and mild head trauma during follow up with families and caretakers of patients with ACs is very important. Additionally, these groups need to receive extra guidance on the potential consequences of rupture, ranging from benign symptoms to subdural hygroma, subdural hematoma, intracystic hemorrhage, and edema with the potential for neurosurgical intervention. Furthermore, the potential for acute neurological symptoms that mimic migraines or strokes needs to be included in the potential consequences of AC rupture, spontaneous or traumatic. 

## 4. Conclusions

Rupture of an arachnoid cyst must be in the differential diagnosis of acute onset of focal neurological symptoms. The symptoms can mimic a stroke or a complex migraine. Patients with arachnoid cysts must be educated about these consequences from rupture of a cyst, and, thus, the potential for stroke-like symptoms should be part of anticipatory counselling in patients with arachnoid cysts. 

## Figures and Tables

**Figure 1 children-08-00078-f001:**
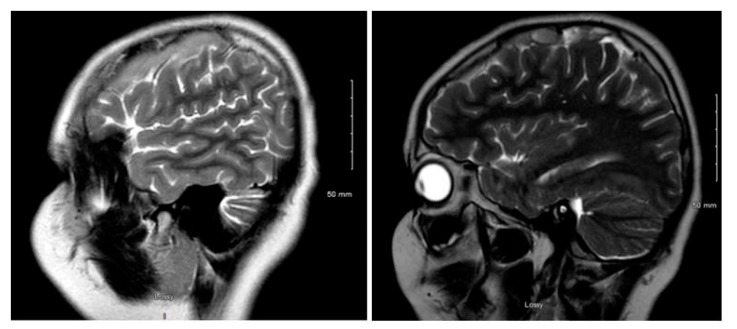
MRI T2WI, sagittal section in 2019 (11 years of age).

**Figure 2 children-08-00078-f002:**
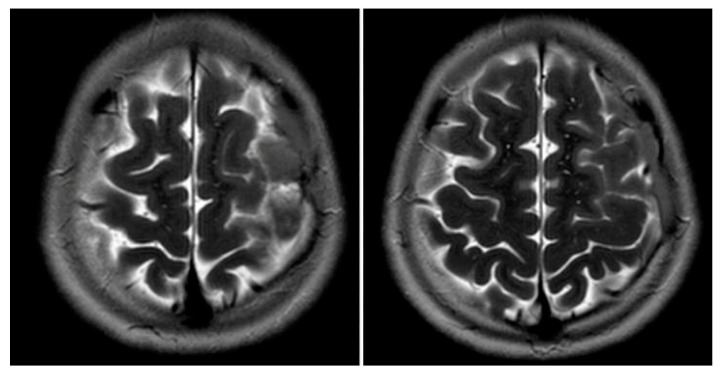
MRI T2WI, axial section in 2019. A very thin subdural hematoma is identified along the left convexity with mild edema noted in the subjacent perirolandic cortex. Decreased T2 signal is now identified within the cyst when compared to Figure 3 and Figure 4, which is suggestive of interval hemorrhage. These findings are consistent with a ruptured arachnoid cyst.

**Figure 3 children-08-00078-f003:**
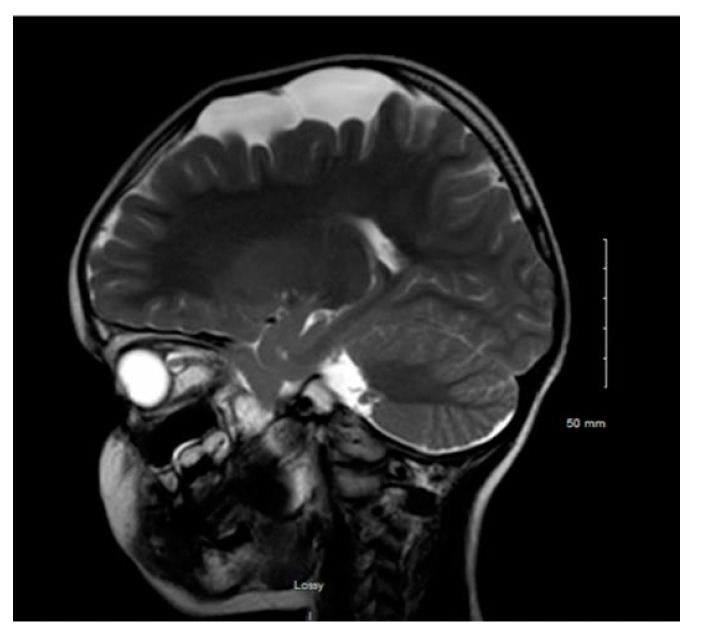
MRI T2WI, sagittal section in 2012 (4 years of age).

**Figure 4 children-08-00078-f004:**
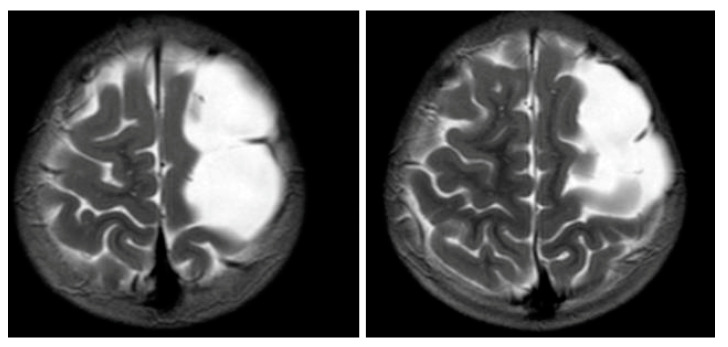
MRI T2WI, axial section in 2012. An extra-axial, well defined CSF structure lesion is identified in the left vertex with the following measurements: anteroposterior, 64 mm; oblique craniocaudal, 53 mm; transverse dimensions, 23 mm. This lesion causes a moderate compression effect of the underlying frontal and parietal gyri at the vertex with scalloping of the inner table. These findings are consistent with a moderate-sized arachnoid cyst at the left frontoparietal vertex.

## Data Availability

The data presented in this study is available upon request from the corresponding author. The data are not publicly available due to privacy of the patient.

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
