# Peer review of "A Rare Case of Spontaneous Arachnoid Cyst Rupture Presenting as Right Hemiplegia and Expressive Aphasia in a Pediatric Patient"

_children, 2021, doi:10.3390/children8020078_

Round 1

Reviewer 1 Report

The study seems to be interesting. However, the paper has to be revised with extinsive modifications. I have given my remarks.

What size do you define as large cyst?

Right facial droop: It can be said that he suffered a facial nerve injury (central facial nerve palsy- the upper part of the face is under normal nerve control, so it is possible to close the eyes and move the forehead muscles, but the facial muscles in the lower part are Facial expression muscles, such as the periphery of the mouth on the other side of the injury, are paralyzed.) Can author comment on this?

Males are four times more likely to develop ACs than females [Ref 5]. Sener R. N. Arachnoid cysts associated with post-traumatic and spontaneous rupture into the subdural space. Only 5 experimenters were involved and this reviewer thinks, that the obtained results are insufficient for a strong conclusion. Can author discuss about this?

Authors mention that "this is the first spontaneous AC rupture that presents with acute onset focal neurological signs. In this case describes a rare spontaneous AC rupture, and as far as we know it is AC rupture-needs a detailed explanation of the neurological signs.

Given the patient's stroke-like presentation, the clinical manifestations of stroke vary widely. I wonder what kind of symptoms you have.

Reviewer 2 Report

The authors present a case of atraumatic arachnoid cyst rupture. The patient was managed conservatively and was discharged two days later.

From the abstract, "He recovered uneventfully and was discharged two days after presentation remaining asymptomatic on subsequent visits. Family with concerns for increased anxiety and mild memory loss since hospitalization."

Clarify that subsequent visits means outpatient visits and not subsequent visits to the hospital. The last sentence is incomplete. "Head and neck MRA were normal" can be removed to meet the word count.

Introduction - line 30-31

Take a second look - "The introduction should briefly place the study in a broad context and highlight why it is important. It shou" can be removed.

It would be worth mentioning an estimation of incidence of spontaneous arachnoid cyst rupture. This should be included. It adds to the novelty of the paper.

Patient description - correct run-on sentences. Include how the cyst was originally discovered.

Line 75 - "MRI imaging" is redundant.

Line 84 - How long is the seizure prophylaxis being continued?

Line 88 - should include a Figure for the follow up MRI if it is being described.

Line 92 - Have you considered the anxiety and memory issues a possible side effect of Keppra?

Line 99 - Again, should include incidence of rupture

Line 105 - it is unclear why there is an emphasis on the "absence of raised intracranial pressure" when there was no intracranial pressure monitored (not indicated) and there is mention of cerebral edema on the MRI description. This could be worded more clearly. 

Line 106 - It sounds like the family does not feel neurological symptoms have ceased. Also, why is there a reference [12] following a sentence described the patient being presented in the current manuscript?

Line 109-110 - run-on sentence. Wouldn't the hemorrhage be more likely to cause this presentation than "protienacious fluid?" There is no discussion of vasospasm...In fact, it is noted that MRA was normal.

Line 116 - How rare?

Line 124 - do you mean cysto-peritoneal shunting or do you mean to include this?

Line 125 - mild head trauma - there is no mention in the physical exam if there were any external signs of trauma for the present case

Line 129-130 - elaborate on the risk factors

Line 137-139 - the last sentence is missing words...

Reviewer 3 Report

The authors do a fine job describing the case of a child who had spontaneous rupture of an arachnoid cyst which resulted in subdural hematohygroma and caused neurologic deficit. The authors could improve this case report by showing in sequence the MRI findings over the period of acute presentation to two weeks later. They describe these findings but don't show them on MRI.

The authors should discuss whether an EEG was done and results. It's very likely the child suffered a seizure due to the ruptured cyst and that resulted in the transient deficits. If an EEG was not done, they should discuss why not.

The authors should go into more detail about the different types of rupture that occur in arachnoid cysts and have they can manifest in young infants to children, teenagers, and adults.

Round 2

Reviewer 1 Report

My remarks are sufficiently addressed.

Reviewer 2 Report

The authors have adequately addressed the issues presented on the initial review.